# Exploring Spatial Patterns in Sensor Data for Humidity, Temperature, and RSSI Measurements

Juan Botero-Valencia [1], Adrian Martinez-Perez [2], Ruber Hernández-García [3,4,*]
and Luis Castano-Londono [5]

1   Grupo Sistemas de Control y Robótica, Faculty of Engineering, Instituto Tecnológico Metropolitano—ITM, Calle 73 No. 76A-354, Medellin 050034, Colombia; juanbotero@itm.edu.co
2   Grupo Materiales Avanzados y Energía, Faculty of Engineering, Instituto Tecnológico Metropolitano—ITM, Calle 73 No. 76A-354, Medellin 050034, Colombia; adrianmartinez@itm.edu.co
3   Research Center for Advanced Studies of Maule (CIEAM), Universidad Católica del Maule, Avenida San Miguel 3605, Talca 3480094, Chile
4   Laboratory of Technological Research in Pattern Recognition (LITRP), Universidad Católica del Maule, Avenida San Miguel 3605, Talca 3480094, Chile
5   Faculty of Engineering, Universidad de Antioquia, Calle 70 No. 52-21, Medellin 050010, Colombia; luis.castanol@udea.edu.co
*   Correspondence: rhernandez@ucm.cl

**Abstract:** The Internet of Things (IoT) is one of the fastest-growing research areas in recent years and is strongly linked to the development of smart cities, smart homes, and factories. IoT can be defined as connecting devices, sensors, and physical objects that can collect and transmit data across a network, enabling increased automation and better decision-making. In several IoT applications, humidity and temperature are some of the most used variables for adjusting system configurations and understanding their performance because they are related to various physical processes, human comfort, manufacturing processes, and 3D printing, among other things. In addition, one of the biggest problems associated with IoT is the excessive production of data, so it is necessary to develop methodologies to optimize the process of collecting information. This work presents a new dataset comprising almost 55 million values of temperature, relative humidity, and RSSI (Received Signal Strength Indicator) collected in two indoor spaces for longer than 3915 h at 10 s intervals. For each experiment, we captured the information from 13 previously calibrated sensors suspended from the ceiling at the same height and with a known relative position. The proposed dataset aims to contribute a benchmark for evaluating indoor temperature and humidity-controlled systems. The collected data allow the validation and improvement of the acquisition process for IoT applications.

**Keywords:** temperature; relative humidity; RSSI; Internet of Things (IoT); indoor climate

## 1. Introduction

The Internet of Things (IoT) is one of the fastest-growing research areas for the development of smart cities, smart homes, and factories. IoT can be defined as connecting devices, sensors, and physical objects that can collect and transmit data across a network, enabling increased automation and better decision-making. In several applications, monitoring humidity and temperature is an essential component for adjusting system configurations and understanding their performance. Both physical parameters are closely related and affect many properties of environments and materials. Humidity measurements are critical for preventing corrosion, condensation, mold, deformation, or other product damage. On the

other hand, temperature monitoring is relevant in storage and testing processes. Some potential consequences of not monitoring humidity and temperature are product damage, inventory loss, unexpected expenses, and equipment failure, among others. Therefore, a wide variety of industries precisely measure these variables, such as healthcare and pharmaceutical [1–4], electronic manufacturing [5], agriculture [6,7], food production [8,9], climate change [10,11], and in the case of enclosed spaces, particularly for the 3D printing process of concrete [12,13], among others. For example, in the pharmaceutical and food industries, there is a high risk of negatively impacting the medication efficacy or biological properties of foodstuffs. Furthermore, improper levels of these variables increase the failure likelihood of equipment in IT environments.

In this context, measurements of these variables in indoor spaces (i.e., buildings, rooms, or warehouses) have acquired great importance in recent years [12,14,15]. For indoor measuring, a correct sampling set-up is crucial to ensure the accuracy of the monitoring system. For instance, measuring points must be representative to avoid over/under-estimations across the space. In addition, repeated measures over time allow for determining variations in conditions and reduce uncertainty due to short-term instrument instability. Hence, the modernization of control systems and sensors has enabled the development of this research area [16,17]. The data collected via these systems have various applications, including physical space analysis [18], biodynamics [19], and virtual sensors [20].

The research areas mentioned above become more relevant for comfort studies in inhabited spaces [12,15,21] and microclimate analysis in warehouses [22,23]. Particularly in indoor areas with a continuous flow of people (e.g., shopping centers, offices, or residential buildings), space conditions can be adjusted depending on the temperature and humidity to optimize environmental comfort. For example, air-conditioning systems, windows, or doors can be automatically controlled to ensure optimal values of these variables. Moreover, there are some places where microclimate variations strongly impact the quality of products, such as the food industry [8], pharmaceutical manufacturing [4], or storage of museum pieces [23]. In these cases, a heterogeneous distribution of temperature or humidity can impact the storage cold chain or the conservation of museum collections. Due to the importance of these application areas, the present study is focused on the analysis of indoor microclimates for IoT applications. IoT device characteristics related to cost, power consumption, weight, dimensions, and connectivity facilitate the deployment of sensor networks for distributed indoor temperature and humidity monitoring. Among other things, IoT applications include 24/7 monitoring of production facilities, pharmaceutical and long-term food storage, data centers, and healthcare environments.

For research purposes, some state-of-the-art (SOTA) works have proposed different data processing algorithms on real-time wireless sensor networks (WSNs). To the best of our knowledge, SOTA approaches are mainly based on classical processing techniques [24,25], machine learning-based models [26–28], and hybrid algorithms [29–31]. Among these studies, one of the most representative databases is the Intel Berkeley Research Lab dataset (IBRL) [32]. IBRL contains unlabelled data collected via 54 Mica2Dot sensors over 37 days within an interval of 31 s. The dataset comprises 2.3 million timestamped registers of temperature, humidity, light, and voltage values. In addition, the authors of [33] introduced the Indoor Temperature and Relative Humidity dataset of controlled and uncontrolled environments. The data were collected for two months at the De La Salle Museum of Natural Sciences and the Laboratory of Control Systems and Robotics, including 4,164,267 values of timestamp, indoor temperature, and relative humidity. The dataset aims to perform studies on processing algorithms at the edge to mitigate drawbacks in real indoor applications. However, neither dataset considers the location of the sensors, which could be useful information for developing methodologies to optimize the process of collecting data. Thus, it can be stated that there is still a lack of public databases that allow a more exhaustive analysis of the performance of the proposed techniques in this area.

The present work introduces a new and the largest dataset comprising almost 55 million records of temperature, relative humidity, and RSSI (Received Signal Strength Indicator)

collected in indoor spaces. The data were collected for nine months (June 2022 to February 2023), over approximately 3915 h at 10 s intervals in two separate indoor places at the Instituto Tecnológico Metropolitano, Medellin, Colombia. We deployed a WSN comprised of thirteen temperature and humidity sensors (Xiaomi Mijia model LYWSDCGQ/01ZM) for data acquisition using a LoPy4 development board. The sensors were previously calibrated and located at the same height suspended from the ceiling, with a known relative position. The proposed dataset aims to contribute a benchmark for evaluating indoor temperature and humidity-controlled systems. The collected data allows the validation and improvement of the acquisition process of IoT applications. The purpose of acquiring this dataset is to allow the development of algorithms to optimize sampling, propose the development of interpolation models, and estimate the influence of humidity or temperature on RSSI. In general terms, our proposal improves how the acquisition of these fundamental variables for human comfort and the development of processes such as 3D printing is done. In addition, it is known that environmental conditions can affect the performance of sensor networks. In particular, temperature could adversely affect the performance of radio transceivers [34]. RSSI information is used in some works to study the effect of environmental conditions on the performance of sensor networks [35–37] or developing algorithms for the estimation of location [38–40]. Considering that the location of the sensors is known, this database can be useful for both types of studies.

The structure of the paper is as follows. Section 2 presents the acquisition devices, collection area, and acquisition process. Section 3 describes the structure of the proposed dataset and analyzes the obtained data. Finally, Section 4 gives the conclusions and outlines future works.

## 2. Materials and Methods

In the following, we present the characteristics of the acquisition devices for deploying the WSN. Moreover, we describe the collection areas and the acquisition process.

### 2.1. Acquisition Devices

Thirteen temperature and humidity sensors Xiaomi Mijia model LYWSDCGQ/01ZM were used for data acquisition using Bluetooth Low Energy (BLE), and a LoPy4 development board was utilized as a single BLE gateway to collect all sensors' data. Each sensor reports the measurements, and the LoPy4 reads the values. Figure 1 depicts the devices used for the deployed WSN.

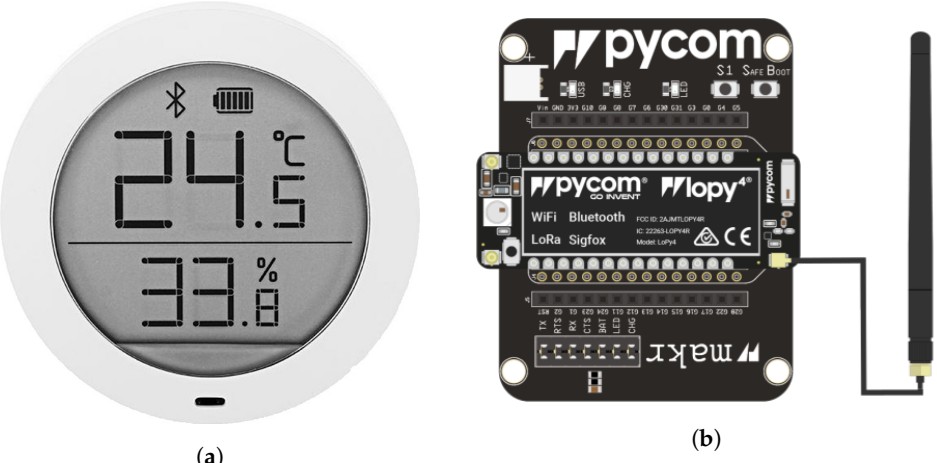

(a)                                                                                                                (b)

**Figure 1.** Devices used for the deployed WSN: (**a**) Xiaomi Mijia Sensor and (**b**) LoPy4 dev board.

The specifications of the temperature and humidity sensors are shown in Table 1. The sensors were calibrated in a Metrology Laboratory of the ITM accredited by the National Accreditation Body of Colombia (Organismo Nacional de Acreditación de Colombia—ONAC). The sensor calibration was performed using a Fluke 2626-H sensor as a reference

measurement instrument. The environmental conditions under which the calibration was performed are between 19.30 °C and 20.50 °C for the temperature and between 56.4%*RH* and 60.5%*RH* for the relative humidity. The calibration range for temperature is from 15° to 40° and for relative humidity from 30%*RH* to 70%*RH*. The specifications of the LoPy4 development board are shown in Table 2.

**Table 1.** Temperature and humidity sensor Xiaomi Mijia model LYWSDCGQ/01ZM specifications.

| Specification | Value |
| --- | --- |
| Temperature range | −9.9 °C–60 °C |
| Temperature accuracy | 0.1 °C |
| Humidity range | 0–99.9%*RH* |
| humidity accuracy | 0.1%*RH* |
| Rated power | 0.18 mW |
| Power supply | Batteries (AAA) × 1 |
| Battery life | 1 year |

**Table 2.** LoPy4 development board specificactions.

| Specification | Value |
| --- | --- |
| Microcontroller | ESP32 |
| RAM | 520 KB + 4 MB |
| External flash | 8 MB |
| Bluetooth | BLE 4.2 and 2.0 |
| Working voltage | 3.3 V to 5 V |

*2.2. Description of Areas*

For data collection, two (2) spaces were selected at the Instituto Tecnológico Metropolitano (ITM) in the Robledo headquarters, located at Calle 73 No. 76A-354 and Fraternidad, located at Calle 54A No. 30-01, both in Medellín, Antioquia, Colombia. These spaces are used by the line of Advanced Computing, Digital Design, and Manufacturing Processes (CADD) of the MATYER research group.

The Modeling laboratory (called LAB1) is located in block F, room 202 of the Robledo headquarters, and is used to carry out simulation studies and the work associated with CADD. The internal dimensions of the room are 7.81 m × 7.82 m and 2.96 m high, as shown in Figure 2a. The 7.82 m side is oriented south–north. On the eastern side, there are two (2) glass windows measuring 2.45 m × 1.80 m and an access door measuring 1.00 m × 2.30 m. The air conditioning equipment is a Comfortfresh model TUB-36CRA-N1 with a 36,000 BTU/h floor–roof fan coil unit located on the south side of the laboratory. The sensor distribution is shown in Figure 2a, and the coordinates are given in Table 3. The vertical installation distance is 0.70 m, measured from the ceiling. In the laboratory, there are two (2) HP XW 6600 workstations, three (3) HP Z600 workstations, eight (8) HP Elite Desks, four (4) Dell Precision T7600, and one (1) Acer Veriton X 4986. There are sixteen (16) luminaires on the ceiling evenly distributed; each one has two (2) 18W LED T8 tubes. Usually, there are five (5) people on site, but in some moments of meetings or training, the number of people may increase to 20.

The simulation, modeling, and prototyping laboratory (called LAB2) is located in the integrated research park of ITM-Fraternidad headquarters. It is used to research simulation and computational processing, among other things. The internal dimensions of the room are 6.60 m × 12.60 m and 4.40 m high, as depicted in Figure 2b. The 12.60 m side is oriented south–north. On the west side, there are five (5) glass windows measuring 1.00 m × 0.50 m, and two (2) access doors of 1.50 m × 2.30 m on the 6.60 m sides. The air conditioning is supplied by two (2) circular grilles of 0.305 m (12 in) diameter, which are connected to a York® brand air handling unit, model YSM-B104V1600CFL0A, with a nominal cooling capacity of 56,000 BTU/h and air supply of 2048 cubic feet per minute (CFM). Figure 2b

shows the sensor distribution, and Table 3 gives their coordinates. The vertical distance of installation measured from the ceiling is 0.60 m. In the space, there are four (4) HP Z600 workstations, four (4) Dell Precision T7600, a domestic refrigerator Challenger model CR074, a proportional hydraulics bench, and a turbine test bench with 10 hp and 2 hp motors. There are ten (10) evenly distributed ceiling lights, each with two (2) 25W T6 fluorescent tubes. Usually, three (3) people stay inside, but for specific academic activities, there can be up to 15 people.

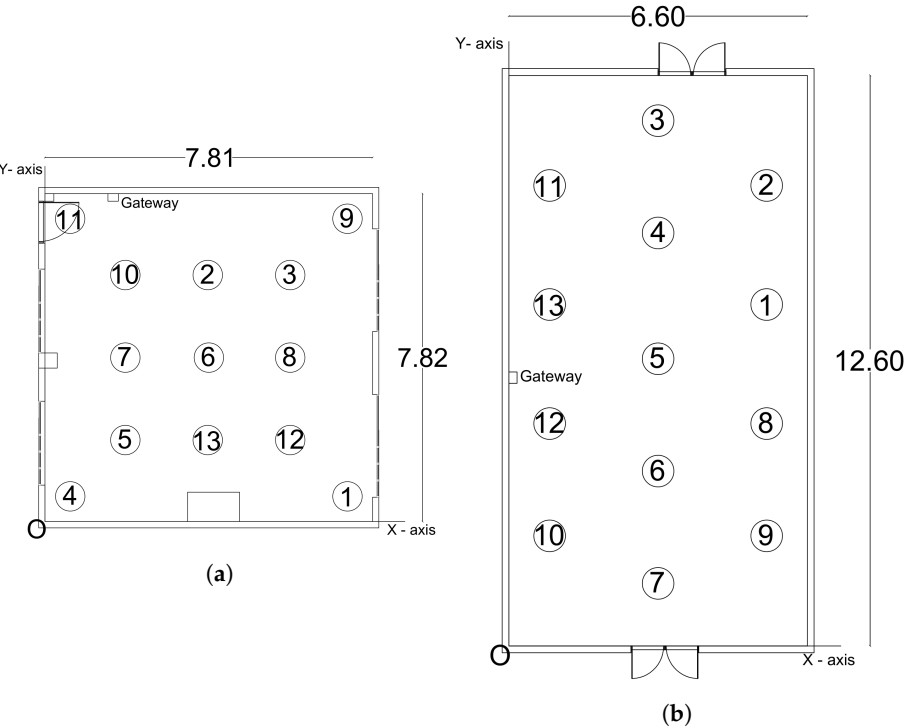

**Figure 2.** Distribution of the sensors for the deployed WSN in (**a**) LAB1 and (**b**) LAB2.

**Table 3.** Coordinates of used sensors and gateway in both laboratories, considering a common origin.

| Sensor | LAB1 | | LAB2 | |
|---|---|---|---|---|
| | X | Y | X | Y |
| 1 | 7.21 | 0.60 | 5.70 | 7.54 |
| 2 | 3.88 | 5.88 | 5.70 | 10.17 |
| 3 | 5.85 | 5.88 | 3.30 | 11.60 |
| 4 | 0.60 | 0.60 | 3.30 | 9.12 |
| 5 | 1.92 | 1.94 | 3.30 | 6.34 |
| 6 | 3.88 | 3.91 | 3.30 | 3.86 |
| 7 | 1.92 | 3.91 | 3.30 | 1.38 |
| 8 | 5.85 | 3.91 | 5.70 | 4.91 |
| 9 | 7.21 | 7.22 | 5.70 | 2.43 |
| 10 | 1.92 | 5.88 | 0.90 | 2.43 |
| 11 | 0.60 | 7.22 | 0.90 | 10.17 |
| 12 | 5.85 | 1.94 | 0.90 | 4.91 |
| 13 | 3.88 | 1.94 | 0.90 | 7.54 |
| Gateway | 1.63 | 7.73 | 0.00 | 5.93 |

## 2.3. Dataset Organization

The dataset is available through the Open Science Framework (OSF) at the following link: osf.io/zbn8w, accessed on 12 April 2023. It consists of 35 files totaling 448.6 MB, organized into seven folders according to the location and date of data acquisition. The file with .mat extension contains the date and time information, as well as the data for temperature,

humidity, and RSSI of the thirteen sensors. The files with `.xlsx` contain the data of the thirteen sensors according to the variable associated with the filename. The date and time values found in the time file are common for the three variables measured, so this information is not included in the files of these variables to avoid redundant data. The dataset organization is presented in Table 4, in which the place, the date, the elapsed time, the size in terms of the amount of data, the filenames, and the corresponding link in OSF can be identified. The Sample06 folder contains the data from the thirteen sensors located close to each other, to be taken as reference data. Finally, in Table 4, the row referenced as EXT corresponds to the temperature and humidity data sampled by an external weather station in a central location of the city, with a 4 min interval and in a time interval that covers all the samplings presented in this article. It should be noted that these data were not totaled, but they can be very useful for analysis of the data presented and correlating them with the external climate.

**Table 4.** Description of the dataset organization.

| Place | Start Date | End Date | Elapsed Time | Size | File | Link |
|---|---|---|---|---|---|---|
| LAB1 | 06/08/2022 10:44:00 | 07/19/2022 11:59:00 | 985:15:00 | 354,695 | Sample00.mat<br>Humidity00.xlsx<br>RSSI00.xlsx<br>Temperature00.xlsx<br>Time00.xlsx | https://osf.io/ra73v<br>https://osf.io/prke9<br>https://osf.io/vbphj<br>https://osf.io/ywv52<br>https://osf.io/3ygax |
| LAB1 | 07/27/2022 11:05:21 | 09/19/2022 12:00:40 | 1296:55:19 | 466,892 | Sample01.mat<br>Humidity01.xlsx<br>RSSI01.xlsx<br>Temperature01.xlsx<br>Time01.xlsx | https://osf.io/te24d<br>https://osf.io/qycu8<br>https://osf.io/f4ce6<br>https://osf.io/vzqr5<br>https://osf.io/x4p27 |
| LAB2 | 10/24/2022 11:00:00 | 10/26/2022 19:00:00 | 56:00:00 | 20,161 | Sample02.mat<br>Humidity02.xlsx<br>RSSI02.xlsx<br>Temperature02.xlsx<br>Time02.xlsx | https://osf.io/qg2ku<br>https://osf.io/3efmz<br>https://osf.io/qvabh<br>https://osf.io/a632n<br>https://osf.io/sed6z |
| LAB2 | 11/03/2022 12:05:00 | 11/27/2022 05:00:00 | 568:55:00 | 204,811 | Sample03.mat<br>Humidity03.xlsx<br>RSSI03.xlsx<br>Temperature03.xlsx<br>Time03.xlsx | https://osf.io/ub8k4<br>https://osf.io/m8e4q<br>https://osf.io/vrjs7<br>https://osf.io/t8wng<br>https://osf.io/hgrbu |
| LAB2 | 11/27/2022 18:30:00 | 11/30/2022 11:20:00 | 64:50:00 | 23,341 | Sample04.mat<br>Humidity04.xlsx<br>RSSI04.xlsx<br>Temperature04.xlsx<br>Time04.xlsx | https://osf.io/s9byf<br>https://osf.io/r8qys<br>https://osf.io/7dbf6<br>https://osf.io/w2vkf<br>https://osf.io/f92eq |
| LAB2 | 12/06/2022 15:15:00 | 01/11/2023 06:46:00 | 855:31:00 | 307,987 | Sample05.mat<br>Humidity05.xlsx<br>RSSI05.xlsx<br>Temperature05.xlsx<br>Time05.xlsx | https://osf.io/cax4m<br>https://osf.io/2pc78<br>https://osf.io/w3x47<br>https://osf.io/h7qkn<br>https://osf.io/undy7 |
| REF | 02/16/2023 15:38:06 | 02/20/2023 07:31:57 | 87:53:51 | 31,644 | Sample06.mat<br>Humidity06.xlsx<br>RSSI06.xlsx<br>Temperature06.xlsx<br>Time06.xlsx | https://osf.io/76zpb<br>https://osf.io/5zrmt<br>https://osf.io/bm6ek<br>https://osf.io/5xz24<br>https://osf.io/8n3bh |
| EXT | 06/08/2022 00:00:00 | 02/20/2023 23:56:00 | 6191:56:00 | 92,880 | Sample07.mat<br>Humidity07.xlsx<br>Temperature07.xlsx<br>Time07.xlsx | https://osf.io/mz3nd<br>https://osf.io/dnj4z<br>https://osf.io/fq4h9<br>https://osf.io/v6jxu |
| | | **Total** | 3915:20:10 | 1,409,531 | | |

### 2.4. Value of the Data

The data obtained in this study were acquired under real operating conditions. Since events occur in the acquisition process that interferes with the continuous acquisition of data, the database is cleaned to provide consistent data for the available periods, avoiding the need to organize or pre-process the data for use. The dataset has a large number of records that allow the development of different types of studies. Estimating indoor temperature and humidity distribution, considering the position of the sensors provides the possibility to study the environmental working conditions in an inhabited environment. In addition, it is possible to include time values to analyze these dynamics over a given period. On the other hand, RSSI data can be used to implement and evaluate position estimation algorithms, taking the known values of the sensor and gateway positions as a reference. For this type of use, the temperature and humidity values could be used to make corrections or improve the positioning algorithms considering the correlation between the measured variables. Furthermore, the data can be studied to establish the effect of temperature, humidity, and position on the performance of the sensor network by analyzing the variation of RSSI as a function of these variables. In this sense, the database can be used by IoT researchers to conduct studies to understand the behavior of sensor networks for monitoring indoor variables.

### 3. Data Samples

Although, as mentioned above, the sensors were calibrated and certified, an experiment was performed in which the sensors were placed in a confined space in its storage box for approximately 87 h. The purpose is that these were close and had the same effect as the environment. The room is air-conditioned (AC) and was used to observe the effect on the measurements under this controlled condition and to determine the correlation between the sensors. Figure 3a,b show the temperature behavior on two different days, in the first case when the temperature varies naturally and in the second where the air conditioning was operated; correspondingly, Figure 3c,d show the same behavior but for humidity on the same days. Figure 3b,d show sudden changes due to the operation of the air conditioning system. These data are included in the repository and may be used by users to make further adjustments to measures if necessary or valid.

Table 5 shows the correlation coefficient matrix for this experiment, showing that in all cases, the correlation coefficient is greater than 98%, even higher than 99% in 70% of cases.

**Table 5.** Coordinates of used sensors and gateway in both laboratories, considering a common origin.

| Sensor | T01 | T02 | T03 | T04 | T05 | T06 | T07 | T08 | T09 | T10 | T11 | T12 | T13 |
|---|---|---|---|---|---|---|---|---|---|---|---|---|---|
| T01 | 100.00 | 98.97 | 98.20 | 99.10 | 99.49 | 99.30 | 98.74 | 99.20 | 99.35 | 99.51 | 98.11 | 99.30 | 99.49 |
| T02 | 98.97 | 100.00 | 99.29 | 99.38 | 99.15 | 99.41 | 99.40 | 99.48 | 99.37 | 98.86 | 98.73 | 99.41 | 98.88 |
| T03 | 98.20 | 99.29 | 100.00 | 99.15 | 98.46 | 98.97 | 99.43 | 99.05 | 98.89 | 98.10 | 99.09 | 98.94 | 98.18 |
| T04 | 99.10 | 99.38 | 99.15 | 100.00 | 99.18 | 99.45 | 99.30 | 99.41 | 99.30 | 99.07 | 99.03 | 99.41 | 99.12 |
| T05 | 99.49 | 99.15 | 98.46 | 99.18 | 100.00 | 99.37 | 98.96 | 99.32 | 99.46 | 99.46 | 98.23 | 99.38 | 99.43 |
| T06 | 99.30 | 99.41 | 98.97 | 99.45 | 99.37 | 100.00 | 99.21 | 99.50 | 99.43 | 99.25 | 98.65 | 99.51 | 99.25 |
| T07 | 98.74 | 99.40 | 99.43 | 99.30 | 98.96 | 99.21 | 100.00 | 99.26 | 99.25 | 98.68 | 99.10 | 99.19 | 98.73 |
| T08 | 99.20 | 99.48 | 99.05 | 99.41 | 99.32 | 99.50 | 99.26 | 100.00 | 99.44 | 99.12 | 98.54 | 99.51 | 99.11 |
| T09 | 99.35 | 99.37 | 98.89 | 99.30 | 99.46 | 99.43 | 99.25 | 99.44 | 100.00 | 99.29 | 98.52 | 99.45 | 99.27 |
| T10 | 99.51 | 98.86 | 98.10 | 99.07 | 99.46 | 99.25 | 98.68 | 99.12 | 99.29 | 100.00 | 98.15 | 99.23 | 99.51 |
| T11 | 98.11 | 98.73 | 99.09 | 99.03 | 98.23 | 98.65 | 99.10 | 98.54 | 98.52 | 98.15 | 100.00 | 98.53 | 98.30 |
| T12 | 99.30 | 99.41 | 98.94 | 99.41 | 99.38 | 99.51 | 99.19 | 99.51 | 99.45 | 99.23 | 98.53 | 100.00 | 99.23 |
| T13 | 99.49 | 98.88 | 98.18 | 99.12 | 99.43 | 99.25 | 98.73 | 99.11 | 99.27 | 99.51 | 98.30 | 99.23 | 100.00 |

In order to demonstrate the practicality of the presented data acquisition, a spatial linear interpolation technique was applied, taking into account the known *x-y* coordinates of each sensor and their consistent height in both experiments. An illustration of this technique can be found in Figure 4, for the LAB1 space, where a specific day was selected,

and interpolation was conducted to compare the temperature, humidity, and RSSI at the time instants of the lowest and highest temperature difference. The outcomes revealed that the area with the most significant temperature variation could be pinpointed and that this region displayed an inverse relationship with the humidity change, as anticipated. The color scale was maintained for both scenarios to facilitate interpretation.

Figure 5 shows the results of the same experiment, but now for space LAB2. In this case, it can be observed that the temperature and humidity have two focal points (Figure 5b,d) because, in this space, there are two air conditioning outlets. In the case of RSSI, the relationship between the actual location of the gateway and the interpolation of the measurements can be observed, where it can be seen that the lower left part of the area where the gateway is located coincides with the sector where the signal is strongest in the graph. The distance represented in *x-y* is in centimeters, and the temperature and humidity respect the original value acquired with the sensors.

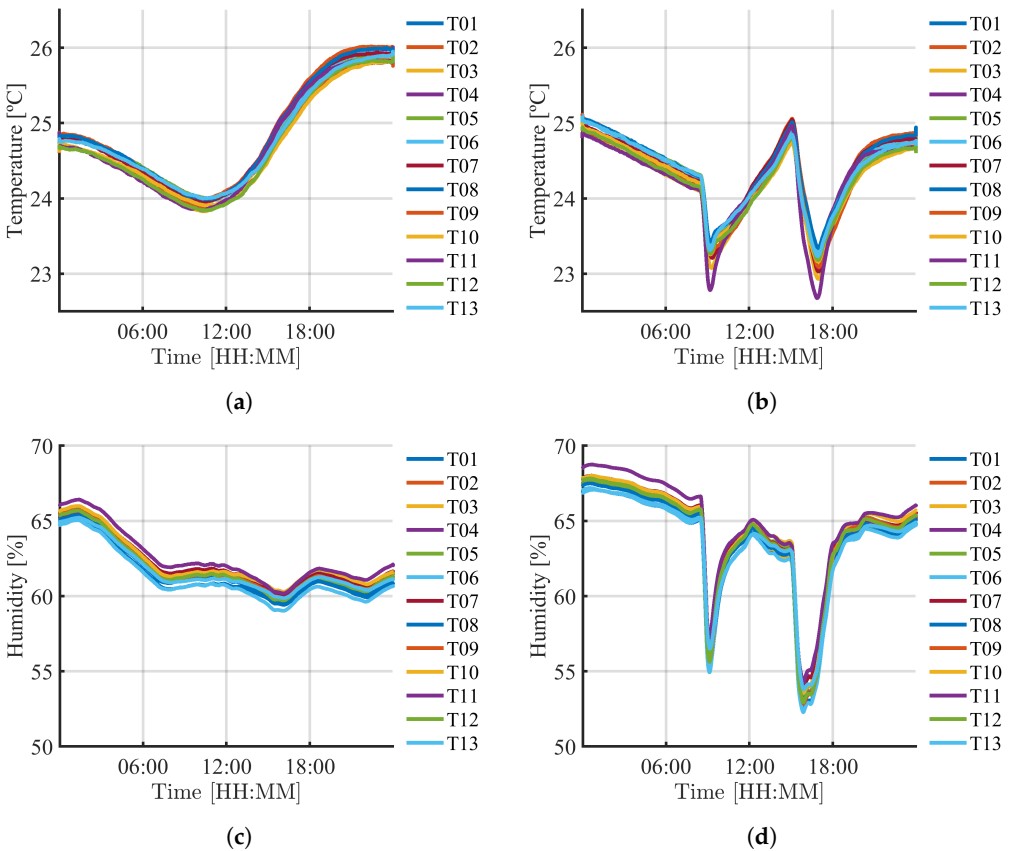

**Figure 3.** Comparative data of temperature and humidity with and without air-conditioned system. (**a**) Temperature without AC; (**b**) Temperature with AC; (**c**) Humidity without AC; (**d**) Humidity with AC.

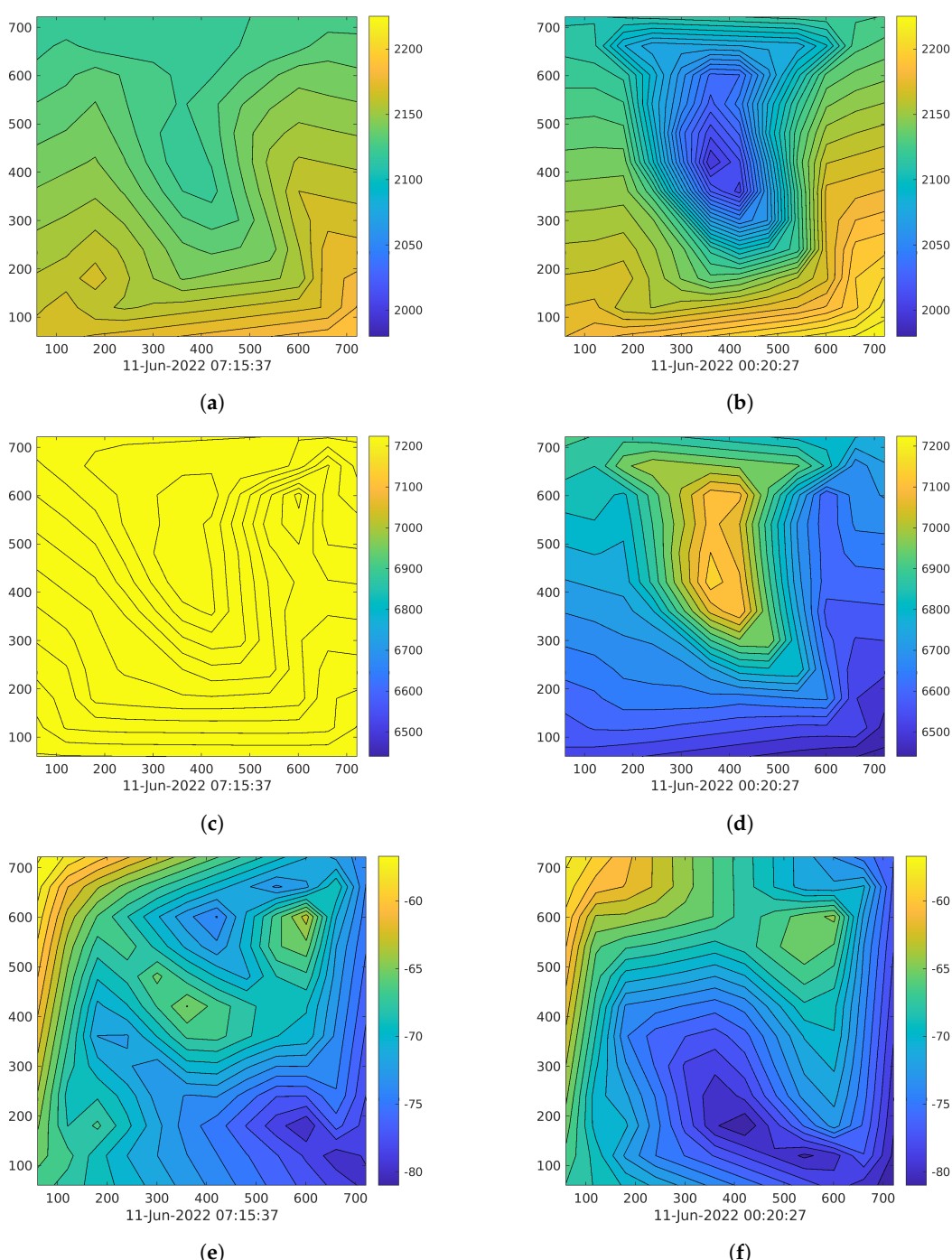

**Figure 4.** Spatial data interpolation for the collected variables in LAB1 at the time of lowest (left column) and highest (right column) temperature difference. (**a**) Temperature interpolation; (**b**) Temperature interpolation; (**c**) Humidity interpolation; (**d**) Humidity interpolation; (**e**) RSSI interpolation; (**f**) RSSI interpolation.

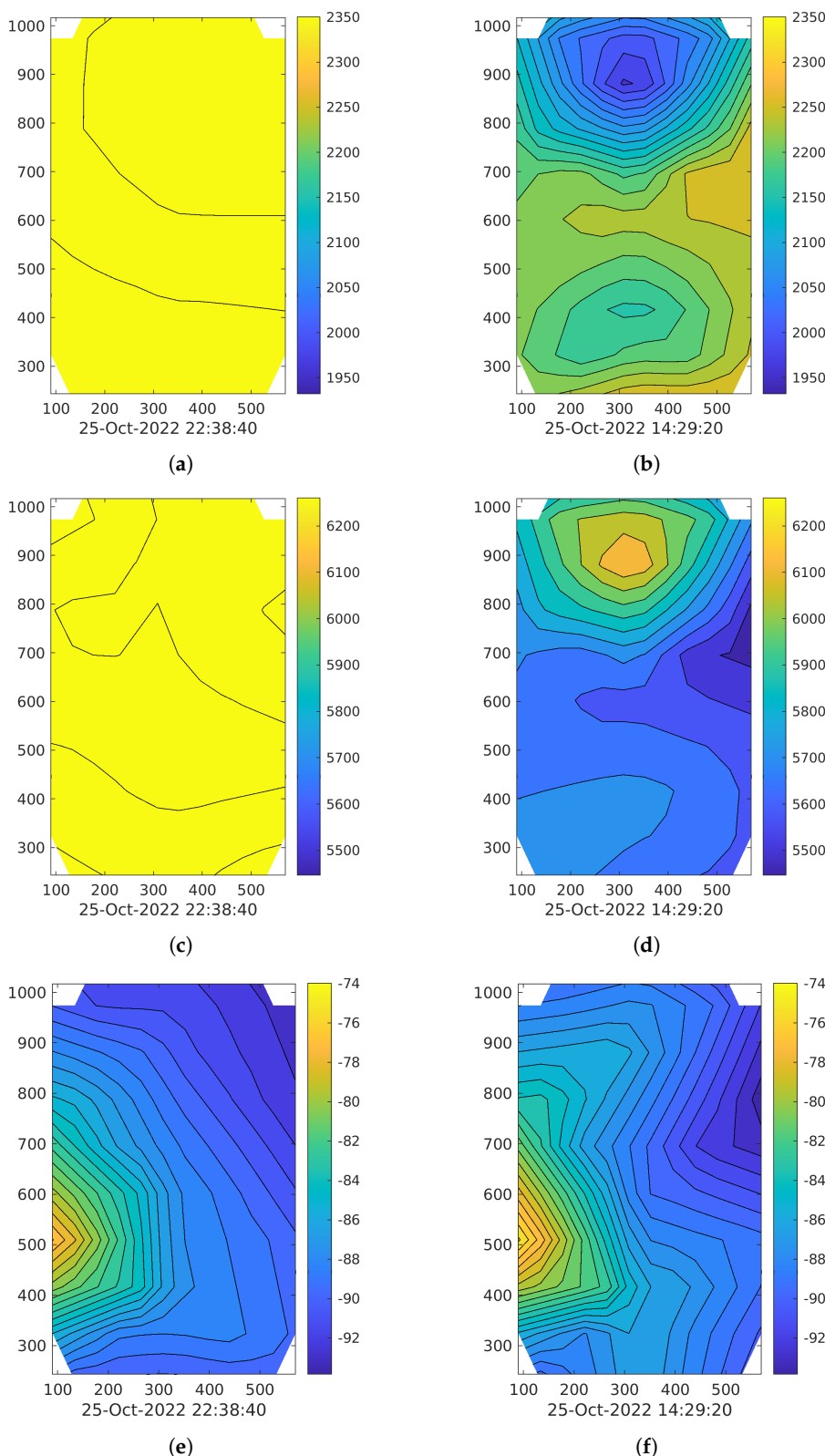

**Figure 5.** Spatial data interpolation for the collected variables in LAB2 at the time of lowest (left column) and highest (right column) temperature difference. (**a**) Temperature interpolation; (**b**) Temperature interpolation; (**c**) Humidity interpolation; (**d**) Humidity interpolation; (**e**) RSSI interpolation; (**f**) RSSI interpolation.

## 4. Conclusions

This paper introduced a new and the largest dataset comprising almost 55 million values of temperature, relative humidity, and RSSI collected over nine months in two separate indoor spaces at the Instituto Tecnológico Metropolitano in Medellín, Colombia. The data collection was performed for nine months (from June 2022 to February 2023), over approximately 3915 h at 10 s intervals. Data are not available for the whole 9-month period of the survey because the database is cleaned to provide consistent data for the available periods, avoiding the need to organize or pre-process the data for use. Thirteen Xiaomi Mijia model LYWSDCGQ/01ZM temperature and humidity sensors and a LoPy4 gateway are used for the data acquisition. The sensors are distributed uniformly in each location and placed at a known distance from the ceiling without contact with any surface. The Xiaomi Mijia sensors underwent calibration at an ITM Metrology Laboratory that holds accreditation from the National Accreditation Body of Colombia (ONAC). The calibration of these sensors involved using a Fluke 2626-H sensor as the reference measurement equipment. This procedure is crucial in guaranteeing the dependability of the collected data.

The collected dataset has some relevant real-world applications in a wide variety of industries that require precise measurement of these variables. Our proposal provides the respective spatial coordinates of the sensors, which allows the interpolation of each measured variable according to the spatial distribution of the sensors. The spatial coordinates are helpful in applying interpolation techniques to estimate variables at other locations within the monitored space. This type of data representation allows for identifying areas with atypical behavior. In the case of air conditioning systems, it allows the analysis of space conditions by observing the gradient that results in space. This information can be useful in the study of thermo-hygrometric modeling in indoor enclosed spaces with air conditioning systems, such as manufacturing plants, warehouses, and data centers. In addition, since RSSI information can be used for position estimation and the location of the sensors is known, this database can be used for the evaluation of RSSI-based localization algorithms or the performance of the sensor network as a function of spatial distribution and environmental conditions. In addition, the proposed methodology can be used for IoT applications, including 24/7 monitoring of production facilities, pharmaceutical and long-term food storage, data centers, and healthcare environments.

**Author Contributions:** Conceptualization, J.B.-V. and A.M.-P.; methodology, J.B.-V. and A.M.-P.; validation, J.B.-V., A.M.-P., R.H.-G. and L.C.-L.; writing—original draft preparation, J.B.-V., A.M.-P., R.H.-G. and L.C.-L.; data curation, J.B.-V.; writing—review and editing, J.B.-V., A.M.-P., R.H.-G. and L.C.-L.; funding acquisition, R.H.-G. and J.B.-V. All authors have read and agreed to the published version of the manuscript.

**Funding:** This research received no external funding. The APC was funded by Universidad Católica del Maule.

**Institutional Review Board Statement:** Not applicable.

**Informed Consent Statement:** Not applicable.

**Data Availability Statement:** Botero-Valencia, J.S., Londoño, L.F.C., Perez, A.F.M., & Hernández-García, R. (2023). Indoor Spatial Patterns Dataset for Humidity, Temperature, and RSSI. DOI: 10.17605/OSF.IO/ZBN8W (accessed on 12 April 2023).

**Acknowledgments:** This study was supported by the Sistemas de Control y Robótica (GSCR) Group COL0123701, at the Sistemas de Control y Robótica Laboratory, attached to the Instituto Tecnológico Metropolitano and the program "Desarrollo de componentes y estructuras bioinspiradas a través de tecnologías de manufactura aditiva para el sector constructor nacional" with code PE22104 and belonging to the second "CONVOCATORIA CONJUNTA DE PROYECTOS DE I+D+i EN EL MARCO DE LA AGENDA REGIONAL DE I+D -> i".

**Conflicts of Interest:** The authors declare no conflict of interest. The funders had no role in the study's design; in the collection, analyses, or interpretation of data; in the writing of the manuscript; or in the decision to publish the results.

## Abbreviations

The following abbreviations are used in this manuscript:

| | |
|---|---|
| IoT | Internet of Things |
| WSN | Wireless Sensor Networks |
| RSSI | Received Signal Strength Indicator |
| BLE | Bluetooth Low Energy |

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
