# Peer review of "Exploring Spatial Patterns in Sensor Data for Humidity, Temperature, and RSSI Measurements"

_data, 2023_

Round 1

Reviewer 1 Report

The dataset is interesting and paper is nicely written. However, I have two minor comments.

1. Add a separate section on the relevance of the data and who is this data useful for?

2. Add a file on corresponding outdoor weather conditions for the measurement location.

Author Response

We thank you for your constructive comments and suggestions that helped us to improve the manuscript. Please, find attached a PDF file responding to the reviewers' remarks.

Looking forward to receiving your favorable consideration.

Kind regards,

Authors of the manuscript

Reviewer 2 Report

The topic is consistent to Data, please answer the following questions:

Introduction:

1. What are some potential consequences of not monitoring humidity and temperature in various industries?

2. How can IoT devices help in the monitoring of temperature and humidity in different applications?

3. What was the purpose of collecting this dataset of temperature, humidity, and RSSI values in indoor spaces?

4. Can you provide more information about the wireless sensor network used for data acquisition, including the type of sensors and development board?

Conclusion:

1. What is the significance of including RSSI information in the dataset of temperature and humidity values collected in the two indoor spaces?

2. How can the spatial coordinates of the sensors be used to interpolate each of the measured variables, and what implications does this have for the study of thermo-hygrometric modeling in indoor enclosed spaces with air conditioning systems?

3. How long was the data collection period for this dataset, and at what intervals were the temperature, humidity, and RSSI values collected?

4. How were the Xiaomi Mijia temperature and humidity sensors calibrated prior to being used for data acquisition?

5. What are some potential real-world applications of the data collected in this study, beyond the evaluation of indoor temperature and humidity-controlled systems?

Minor editing of English language required

Author Response

(The authors gave the same response as above.)
